# Farmer Evaluation of Irrigation Services. Collective or Self-Supplied?

Laura Mirra [1], Bernardo Corrado de Gennaro [2] and Giacomo Giannoccaro [2,*]

1   Department of Agricultural Science, University of Naples Federico II, 80138 Naples, Italy; laura.mirra@unina.it
2   Department of Agricultural and Environmental Science, University of Bari Aldo Moro, 70121 Bari, Italy; bernardocorrado.degennaro@uniba.it
*   Correspondence: giacomo.giannoccaro@uniba.it; Tel.:+39-080544-2885

**Abstract:** Economic evaluation of farmland is an important issue in the agricultural sector. The aim of this study was to quantify the economic value of land in the farmland area of the Reclamation and Irrigation Board of Capitanata (Apulia region), differentiating by irrigation water service type (collective or self-supplied). The analysis involved a heuristic evaluation using the hedonic pricing method of the sales comparison approach. The data was gathered through a survey on a group of 75 farmers. The results showed higher capitalization values in the case of lands served by self-supplied sources from groundwater. Actually, in the long-term, an enhanced reliability was found for the self-supplied rather than collective services. The findings highlight the importance for collective water associations of differentiating water rights allocations on the basis of a volume guarantee. In future, water user associations of collective services could consider a different water right system along with price discrimination to efficiently allocate the resource among farmers and improve the sustainability of current water management.

**Keywords:** irrigation water service; water value; hedonic pricing method

## 1. Introduction

The economic evaluation of water resources plays a pivotal role in agricultural sectors in which the water market does not function properly. Even in those countries where water markets have long been functioning properly (i.e., Australia, Chile and the United States), prices and water value are often influenced by government intervention (e.g., buy-back schemes to meet environmental standards). Assessment of the economic value, at least for agricultural, urban and industrial uses, is the corner stone of the Water Framework Directive (60/2000/EC) in the European Union. Information on the economic value should support policy makers in defining effective pricing policies and in recovering all associate costs, which in turn should lead to an efficient allocation of water resources [1].

Despite being relevant, valuing irrigation water can be very challenging due to the lack of water accountability in agriculture [2], mostly in cases of on-farm groundwater wells [3]. Appraisal methods for determining economic value of irrigation water, their characteristics and uses, are well reported in [4] (pp. 44–45).

Variations in the quantity of water available, its quality [5] and the timing and location of supplies can lead to significant adjustments in the usage value of water for irrigation. Berbel et al. [1] found crop differences for irrigation water value, supply security being one of the emerging features in intensive agriculture systems [6], even more so under scarcity conditions, such as hydrological drought [7]. While the economic benefits of irrigation water have been largely investigated, neglecting long-run water services functioning, little attention has been paid to the type of irrigation water services. It is worth mentioning that the reliability of supply services in the contest of precision farming is emerging as



very relevant issue in irrigation. Advantages attached to the diffusion of smart irrigation technologies might be frustrated by unreliable irrigation services.

Broadly speaking, irrigation water services can be either collective or self-supplied. In the case of irrigation water supplied by collective networks, farmers can rely on the irrigation infrastructures of pipelines conveying water under pressure (either in rotation or on demand) at the plot gate. Water supply is managed by a local irrigation board, generally known as the water user association. The water source is generally diverted from river basins which are regulated by means of dam systems. The extent of irrigated land varies according to the reservoir capacity, while the available amount of water supply depends on the rainfall and snowfall pattern of previous seasons. Water user associations are in charge of defining water rights for irrigation; i.e., how much water farmers can seasonally benefit from, when they can do so and a priority rule (if there is one) [8]. They also establish the water pricing policy. With rare exceptions (e.g., Australia), the usage rights are attached to the land and depend on the presence of infrastructures, and they cannot be sold separately (neither temporarily nor permanently).

In the case of non-networked water supplies, irrigation services are self-organized by the end users. Irrigation water can be directly diverted by farmers from rivers and withdrawn from groundwater aquifers as well. The latter case is undoubtedly the most common water resource when irrigation services are self-supplied. Although drilling private wells is generally subject to public authorization or licensing [9], among other issues, the high cost incurred by public authorities for monitoring and controlling groundwater use might prevent its sustainable management [10]. Regarding non-regulated surface and groundwater services, end users pay for all the financial costs of the water supply. In addition to fees for licensing, access to water sources can be charged with environmental taxes, as documented in some European Member States [11]. In the case of self-supplied services based on groundwater resources, farmers might face the problem of over-exploitation. Although the groundwater source is generally considered to be unlimited [12], overextraction from aquifers can lower the groundwater table, which in turn increases the pumping cost of irrigation services. Likewise, as a consequence of excessive groundwater withdrawal along the coastline, increasing salt intrusion reduces water quality.

In this context, this study analyzed the economic value that farmers place on irrigation water services, namely collective and self-supplied water services. Basically, irrigation allows the growing of cash crops, which give higher returns to farmers than the ordinary rain-fed crops, increase yields and raise the productivity of agricultural activity. A further point of interest is that irrigation performance can be also influenced by the type of water service. The research question relates to supply uncertainty and the economic value of irrigation water services in the long run. So far, self-supplied services from groundwater have been assumed to be more flexible, accessible and cheaper than collective managed supply services (see, for instance, [12,13]. As a consequence, in the mainstream literature concerning water management for irrigation, command and control policy implications arise (e.g., restriction to groundwater access, metering and rise in price for limiting volume withdrawal). Although groundwater in many regions is still easily accessible and affordable, the main advantages farmers can take relate to lower uncertainty of water supply. The magnitude of this issue clearly emerges in those regions where conjunctive use of surface and groundwater resources takes place [12].

In this research we checked whether and how much the land market can reveal the actual value of supply reliability of services for irrigation. A heuristic evaluation of irrigation water was carried out using the hedonic pricing method of the sales comparison approach.

A comparison between irrigated and rain-fed land sales value was performed. The analysis was based on an explorative survey conducted among 75 farmers in the Capitanata area, the largest irrigated area of Apulia region (Southern Italy). With reference to the Italian context, almost 65% of irrigated land is served by a collective pipeline network. Nevertheless, the Apulia region shows a different pattern, with self-supplied services from groundwater being the most frequent (almost 70% of irrigated land) [14]. The area was

chosen as explorative case study, in which conjunctive use of surface and groundwater resources has been active for a long time [12].

Based on the findings, policy implications are proposed both to policy makers and to board advisors of collective irrigation facilities. For the latter, there should be room for raising the economic value of irrigation services, moving away from the actual central planning allocation rule. While the proportional rule is the most widely used rationing method to allocate water in cases of water scarcity, according to the results the implementation of security-differentiated water rights is advisable.

Although the case study was used to check the hypothesis that long-term reliability of water services is already capitalized in the land value, the findings are not limited to the study area. Therefore, the policy implications can be extended to broader national and international contexts.

## 2. Materials and Methods

### 2.1. Sales Comparison Approach: Conceptual Framework

There are many scientific analyses that use the hedonic price method. The basic assumption of the method is that the market price of a good or service is a function of its characteristics, and that an implicit price exists for each of the characteristics. This method has been applied to assess a variety of goods, such as cassava-wheat composite bread [15] or bottled water [16].

In the case of the land market, results of hedonic price analyses are reported both for developed countries [5,13] and developing ones [17]. A typical approach to the topic would begin with a model in which the reported selling price of farmland is regressed against a number of explanatory variables, such as size of plots, distance to nearest town and so on. Moreover, in order to isolate irrigation effects, the amount of water rights is used as predictor [18]. This formal statistical analysis of farm sales prices needs a consistent number of observations on the land markets. Moreover, since the a priori form is generally not known for econometric regression, model calibration is based on statistical goodness and the researcher's skills. Nevertheless, as reported by Mallios [19] on the same sample data, different regression models can be performed with high goodness. Yet, in the parametric regressions, the irrelevance of collinearity among the explanatory variables is normally assumed. In fact, this assumption is not always proven for land characteristics [5].

In light of those shortcomings, in this study the sales comparison approach, as proposed by Young (2005) [18], was applied. This method belongs to the hedonic pricing model category. It is based on determining the differences in price for similar land, except for the water resource. The difference between the prices of irrigated and non-irrigated land represents the sales value ($V_i$) while $B_i$ is the yearly rent value obtained from irrigation. In addition, if the monetary values can be related to the actual volume usage for each irrigated piece of land, it is possible to obtain an average monetary value per volume ($€/m^3$) [20]. In the absence of the water quantity for the individual sales observations, the approach is termed "quasi-hedonic" [18,21]. Equation (1) leads to the capitalization value. $r$ represents a rate of discount at which entrepreneurs expect to capitalize a future stream of rents from irrigation. Generally, the higher the rate, the more uncertainty is associated with the stream of gains. Conversely, the Equations (2) and (3) illustrate how to obtain the annual lease equivalent (rental price) and the capitalization rate, respectively.

$$V_i = B_i/r \qquad (1)$$

$$B_i = V_i \ x \ r \qquad (2)$$

$$r = B_i/V_i \qquad (3)$$

For the validity of the method, it is important that the land sales comparison occurs between observations with similar characteristics, except for the water resource. Only in this way can the real contribution of the water be isolated. For this purpose, a *cluster analysis* was carried out to identify homogeneous groups of observations. Cluster analysis is a class

of techniques used to classify objects or cases into corresponding groups called clusters. These techniques can be grouped into two macro-categories: hierarchical agglomerative methods and iterative partitioning methods. In this study, in order to identify homogenous groups of observations, a hierarchical agglomerative method was used.

Within each homogeneous group, observations can be compared to others with similar characteristics.

The first step of the method consists of forming a matrix which represents the pairwise similarities of all objects being clustered. Subsequently, according to a specific algorithm, the method proceeds to gradually build clusters by merging the most similar objects together at each step. The final output can be represented by hierarchical trees or dendrograms [22].

In order to group farm observations, Ward's method was applied here [23]. By means of this method, groups of homogenous data are identified, minimizing variance within clusters and maximizing variance between clusters. To measure the distance between elements, the Euclidean distance was used. The variables used as determinants for group observations are listed in Table 1. We assumed that the differences between observations depended on farms' structural characteristics (land size, irrigated crop share, labor and water availability) and farmer characteristics (age, credit access and off-farm job). Hence, we considered all variables listed in Table 1 to calibrate the cluster. A strong element of innovation compared to the pre-existing literature is the variable that considers the type of water service (self-supplied or collective service) adopted by farmers. Clustering was performed using SPSS Statistics 26 software.

**Table 1.** Variables used as determinants in cluster analysis.

| Variable | Type | Code | Sample | |
|---|---|---|---|---|
| *Farm characteristics* | | | Mean | SD |
| Total land owned | Metric | ha | 43.92 | 57.87 |
| Managed irrigated land | Metric | ha | 61.16 | 75.27 |
| Irrigated land rent-in | Binary | no = 0; yes = 1 | 29.25 | 44.70 |
| Family workers | Binary | no = 0; yes = 1 | 0.65 | 0.48 |
| Extra-family workers | Binary | no = 0; yes = 1 | 0.14 | 0.35 |
| *Irrigated crop pattern* | | | | |
| Tomato | | | 11.79 | 25.84 |
| Vineyards | | | 25.01 | 37.39 |
| Vegetables | | | 16.49 | 32.28 |
| Permanent vegetables | | | 6.89 | 16.73 |
| Olive grove | Metric | crop pattern (%) | 8.51 | 22.11 |
| Intensive olive grove | | | 9.70 | 25.26 |
| Orchards | | | 5 | 14.75 |
| Cereals | | | 4.70 | 14.44 |
| Others | | | 9.70 | 24.04 |
| *Water* | | | | |
| Multiple water services | Binary | 0 = single; 1 = multiple | 0.18 | 0.38 |
| On-farm metering device | Binary | no = 0; yes = 1 | 0.15 | 0.35 |
| Innovation in irrigation field [1] | Binary | no = 0; yes = 1 | 0.14 | 0.34 |
| *Farmer characteristics* | | | | |
| Age | Metric | years | 50.18 | 11 |
| Credit access | Binary | no = 0; yes = 1 | 0.46 | 0.50 |
| Off-farm job | Binary | no = 0; yes = 1 | 0.21 | 0.41 |

[1] Digitalized irrigation management systems. Source: direct survey.

## 2.2. Data Sources

The case study was located in the south of Italy (Apulia) within the administrated area of the Reclamation and Irrigation Board of Capitanata (Province of Foggia) (Figure 1). The

case study concerned a highly developed agricultural area, which is the largest irrigated area of Apulia. The consortium extends over 441,579 ha of administrative area, of which 140,378 ha is served by a collective irrigation network, namely the Fortore and Sinistra Ofanto schemes, both equipped with modern, pressurized, on-demand delivery services. A mixed payment scheme is used: (i) a yearly fixed component related to the farm irrigated area (60 €/ha); (ii) a volumetric three-tier water tariff scheme based on irrigation water use. Average annual volume delivered by collective services amounts to 103 Mm$^3$ but reliability of irrigation services is reduced by periodic water shortages [12,24]. In such cases, the proportional method based on all farmers being assigned an amount (water allocation) proportional to their claim (ultimately, the extent of farmland) is applied. The rationing method allocates the same volume of water irrespective of crop irrigation requirements, neither crop value nor economic losses.

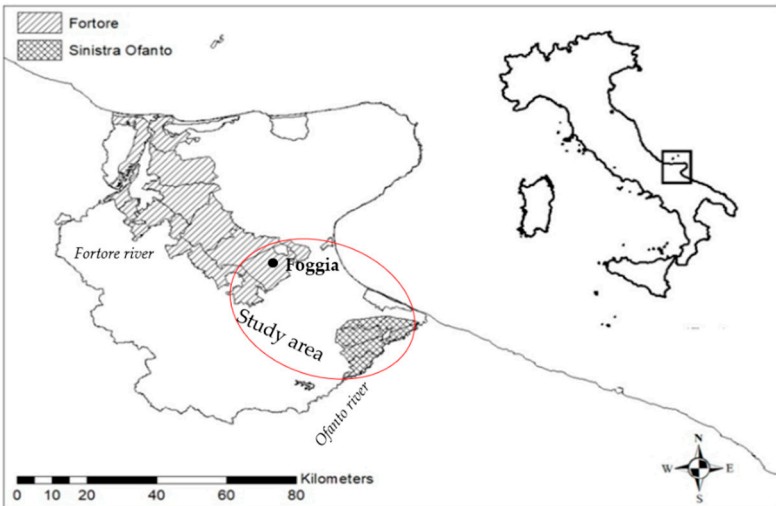

**Figure 1.** Study area. Source: Adapted with permission from Giannoccaro [24].

Overall annual irrigated land accounts for 121,266 ha, of which almost 50% is mostly supplied by collective irrigation services. The remainder is irrigated directly by self-supplied services, normally from groundwater resources. In several cases, farmers rely on both types of services. Vineyards, tomato processing and fresh-cut vegetables are the most profitable irrigated crops grown in the area.

A sample dataset of 75 observations was obtained through a face-to-face survey of farmers carried out in 2019 by trained interviewers. A snowball sampling procedure was followed in order to align sampled cases as much as possible to irrigated crop pattern, annual water used and irrigation services type in the study area. Sampling was conducted in a homogenous area with relevant cases of conjunctive use of both irrigation services (see red circle in the Figure 1). Despite the size of the sample, a higher proportion of farmland area of the study area was sampled, namely 4861 hectares of total utilized agricultural area with 2017 ha actually irrigated. As a whole, the sample showed a good representativeness of irrigated crops (Table 2) and water service type (Table 3).

A structured questionnaire was administrated to farmers in order to gather data on farm cropping patterns, soil fertility, the distance from the nearest town and road, the slope and size of each farm plot, structural and sociodemographic farmer characteristics, irrigation water use and water resource management. The total number of observations for irrigation was 69.

**Table 2.** Sample representativeness.

|  | Sample |  | Study Area |  |
|---|---|---|---|---|
| Total farmland (ha) | 4861 |  | 237,951 |  |
| Irrigated land (ha) | 2017 |  | 66,536 |  |
| Yearly average irrigation volume (m$^3$/ha) | 2386 |  | 2740 |  |
| *Irrigated crops land:* | *(ha)* | *%* | *(ha)* | *%* |
| Tomato | 363 | 18 | 13,442 | 20 |
| Vineyards | 323 | 16 | 22,312 | 33 |
| Orchards | 82 | 5 | 3396 | 5 |
| Vegetables | 741 | 37 | 11,331 | 17 |
| Permanent vegetables | 176 | 8 | 1577 | 2 |
| Olive grove | 150 | 7 | 13,089 | 20 |
| Others | 182 | 9 | 1389 | 3 |
| Total | 2017 | 100 | 66,536 | 100 |

Source: sample data from direct survey and study area adapted with permission from Giannoccaro [24].

**Table 3.** Crop pattern by water service.

| Water Service | Collective |  | Self-Supplied |  | Multiple Service |  |
|---|---|---|---|---|---|---|
| N | 19 |  | 37 |  | 13 |  |
| SAU (ha) | Total | Irrigated | Total | Irrigated | Total | Irrigated |
| Total | 1149.75 | 565.17 | 1729.09 | 735.89 | 1.983 | 716.50 |
| Cereals | 501.71 | 56.7 | 862.78 | 26.9 | 1253 | 46 |
| Legumes | 142.15 | - | 185.51 | - | 100.5 | 12 |
| Tomato processing | 74.31 | 74.31 | 50 | 50 | 259 | 259 |
| Fresh tomatoes | - | - | 3.5 | 3.5 | - | - |
| Potatoes | - | - | - | - | 3 | 3 |
| Melon | - | - | - | - | 5 | 5 |
| Cabbage | 50 | 50 | 156 | 156 | 101 | 101 |
| Beet | 238 | 238 | 75 | 75 |  |  |
| Lettuce | - | - | - | - | 6 | 6 |
| Fennel | - | - | 2 | 2 | 104 | 104 |
| Artichoke | - | - | 30 | 30 | - | - |
| Asparagus | 35.66 | 35.66 | 50 | 50 | 61 | 61 |
| Secular olive grove | 5.5 | 4.5 | 3.5 | 3.5 | - | - |
| Olive grove | 4.2 | 4.2 | 69 | 58 | 28.5 | 18.5 |
| Intensive olive grove | 28.5 | 28.5 | 14.99 | 14.99 | 22.5 | 22.5 |
| Trellised vineyards | 3 | 3 | 16.5 | 16.5 | 30 | 30 |
| Tent vineyards | 32.8 | 32.8 | 137.01 | 137.01 | 10.5 | 10.5 |
| Table vineyards | 33.5 | 33.5 | 18.8 | 18.8 | - | - |
| Orchard fruits | 4 | 4 | 62 | 62 | - | - |
| Others | 2.7 |  | 61.5 | 32.5 | 76 | 76 |

Source: direct survey.

During the interview, respondents were asked to indicate the price of renting and purchasing land of owned estates either for irrigated or non-irrigated land, respectively. Generally, for the same land plot, real estate values of both leasing and selling prices were not available. In order to apply Equation (3), this piece of information was very relevant. While the usual approach is to try to approximate the discount rate for long-term loans for land purchases, in this research, $r$ was obtained from survey data. Since plot differences can exist within the same farm, respondents referred the stated value to the most economically significant crop. Farmers were also asked whether they had bought/sold or leased/rented land in the past five years. Indeed, 32 farmers stated that they actually bought some land in the last five years and 22 had leased it. These figures supported the land price information gathered.

Furthermore, with regard to the self-supplied water service, farmers were asked if they had encountered problems with the quantity or quality (salinity) of water. Although the service is characterized by a high degree of reliability, it is not exempt from limitations. A total of 30 farmers indicated quantity problems while only three had quality (salt-related) issues. Finally, respondents were asked to indicate the running costs they incurred annually for self-supplied irrigation services.

## 3. Results

### 3.1. Homogeneous Farmer Groups

The cluster analysis identified four groups differing in terms of farm structural characteristics (land and labor endowment, irrigated crop pattern, water-related information) and farmer characteristics (Table 4).

The first cluster identified medium–large farms with an average of 13.69 ha of irrigated land, the lowest size among the groups. No hired labor was indicated. The irrigated crop pattern was characterized by olive groves. None relied on multiple water services. This was also the oldest-aged farmer group, with 31% having off-farm jobs.

The second cluster was the most representative of the study area, including 37 farms. The farms belonging to this group were small in size, with an average of 28 hectares owned. The main crops grown were vineyards, with 44.68% of farm hectares. Only three observations out of 37 indicated multiple irrigation services. There was a very small percentage of farmers who had adopted irrigation-related innovation (16%) in the past five-year-period. The number of those having off-farm jobs was still significant.

The third type of farm identified included 12 observations and larger farm sizes (on average 39.21 hectares owned). Also, the amount of managed irrigated land increases up to 38.55 hectares. A greater flexibility in expanding farmland size was found, with 75% of responses indicating rent-in land. Labor hiring was also significant. The irrigated crop pattern differed significantly from the previous groups: tree crops were almost absent in favor of tomato processing (58.81%) and permanent vegetables, such as artichokes and asparagus (21.31%).

Multiple water services grew sharply, with 58% of farms having such an option. The average farmer age decreased, while the percentage of those with access to credit increased. Those having an off-farm job were totally absent.

Finally, the fourth cluster identified was characterized by the largest farm size. As a consequence, the irrigated hectares of the entire sample were concentrated in this group. The mean values of the total land owned and the managed irrigated land amounted to 185 and 154.5 ha, respectively. The values of managed irrigated land and extra-family workers were the same as for cluster 3. The main irrigated crops were vegetables (fresh-cut crops, 46%) and tomato processing (25.13%). This group showed the highest values for the percentages of farmers who used multiple water service (75%) and measured the water resource (55%). Moreover, the average farmer age was 41 and all farms adopted innovations in the irrigation field. Finally, no farmer had an off-farm job.

**Table 4.** Cluster features.

| | Cluster 1 | | Cluster 2 | | Cluster 3 | | Cluster 4 | | Sample | |
|---|---|---|---|---|---|---|---|---|---|---|
| **N** | **16** | | **37** | | **12** | | **4** | | **69** | |
| | **Mean** | **SD** | **Mean** | **SD** | **Mean** | **SD** | **Mean** | **SD** | **Mean** | **SD** |
| *Farm characteristics* | | | | | | | | | | |
| Total land owned | 47.42 | 59.20 | 28.53 | 24.14 | 39.21 | 35.37 | 185 | 135.27 | 43.93 | 57.87 |
| Managed irrigated land | 13.69 | 15.59 | 20.23 | 18.11 | 38.55 | 31.77 | 154.5 | 115.18 | 29.25 | 44.70 |
| Irrigated land rent-in | 0.12 | 0.35 | 0.16 | 0.37 | 0.75 | 0.45 | 0.75 | 0.5 | 0.28 | 0.45 |
| Extra-farm workers | 0 | 0 | 0.16 | 0.37 | 0.25 | 0.45 | 0.25 | 0.5 | 0.14 | 0.35 |
| *Irrigated crop share* | | | | | | | | | | |
| Tomato | 4.23 | 11.57 | 1.12 | 4.20 | 50.81 | 37.13 | 23.63 | 25.13 | 11.62 | 25.19 |
| Vineyards | 4.37 | 13.15 | 44.68 | 41.33 | 0.23 | 0.80 | 0 | 0 | 24.65 | 37.24 |
| Vegetable crops | 0 | 0 | 22.72 | 38.45 | 11.61 | 18.57 | 39.49 | 43.36 | 16.26 | 32.10 |
| Permanent vegetable crops | 1.86 | 7.43 | 4.35 | 9.74 | 21.31 | 32.20 | 7.32 | 8.93 | 6.79 | 17.73 |
| Olive grove | 26.14 | 39.12 | 3.82 | 10.05 | 1.53 | 2.92 | 2.27 | 4.45 | 8.39 | 21.97 |
| Intensive olive grove | 25.95 | 42.31 | 5.72 | 16.00 | 0 | 0 | 10.71 | 21.42 | 9.56 | 25.10 |
| Orchards | 0 | 0 | 9.31 | 19.22 | 0 | 0 | 0 | 0 | 4.92 | 14.65 |
| Cereals | 8.26 | 22.58 | 2.71 | 9.92 | 7.68 | 14.88 | 0 | 0 | 4.70 | 14.44 |
| Other | 29.17 | 39.56 | 1.14 | 5.43 | 7.09 | 17.59 | 18.83 | 26.94 | 9.56 | 23.89 |
| *Water resource* | | | | | | | | | | |
| Water service type | 1.5 | 0.51 | 1.89 | 0.51 | 2.33 | 0.88 | 2.5 | 1 | 1.91 | 0.67 |
| Multiple water services | 0 | 0 | 0.08 | 0.27 | 0.58 | 0.51 | 0.75 | 0.5 | 0.18 | 0.39 |
| Water accounting | 0.06 | 0.25 | 0.13 | 0.34 | 0.25 | 0.45 | 0.55 | 0.58 | 0.16 | 0.37 |
| *Farmer characteristics* | | | | | | | | | | |
| Age | 55.69 | 10.76 | 51.93 | 9.43 | 41.83 | 8.18 | 41.75 | 12.5 | 50.3 | 10.91 |
| Credit | 0.19 | 0.40 | 0.43 | 0.50 | 0.83 | 0.38 | 0.75 | 0.5 | 0.45 | 0.50 |
| Innovation | 0 | 0 | 0.16 | 0.37 | 0 | 0 | 1 | 0 | 0.14 | 0.35 |
| Off-farm | 0.31 | 0.48 | 0.27 | 0.45 | 0 | 0 | 0 | 0 | 0.21 | 0.41 |

Source: our elaboration from the direct survey.

It is possible to identify two macro-categories based on the entrepreneurial figure. The first macro-category, consisting of clusters 1 and 2, included farmers whose utility function was associable with the maximization of net income. The size, the irrigated crop pattern and the labor availability did not require high managerial skills nor a sufficient profitability as an exclusive activity. There was a high level of personal endowment for labour and capital. The second macro-category, however, consisting of clusters 3 and 4, included farmers whose utility function could be associated with profit maximization. The size, the crop-pattern, the acquisition of land for rent, the access to credit and the age and off-farm job work were indicators of dynamism and flexibility, typical characteristics of a professional business management system [25].

Crop patterns, especially irrigated corps, were the main difference that cluster analysis reported. Net income maximizers exhibited smaller farmland size and were basically specialized in permanent crops (i.e., orchards, vineyards and olive groves). By contrast, profit maximizers relied on larger farmland size and were oriented toward arable crops (i.e., vegetables and tomato processing). A further important differentiation between the two macro-categories was the type of water service adopted. In particular, the net income

maximizers almost exclusively indicated a single water service (collective or self-supplied); conversely, profit maximizers relied on multiple irrigation services.

A graphic representation of the cluster results is shown in Figure 2.

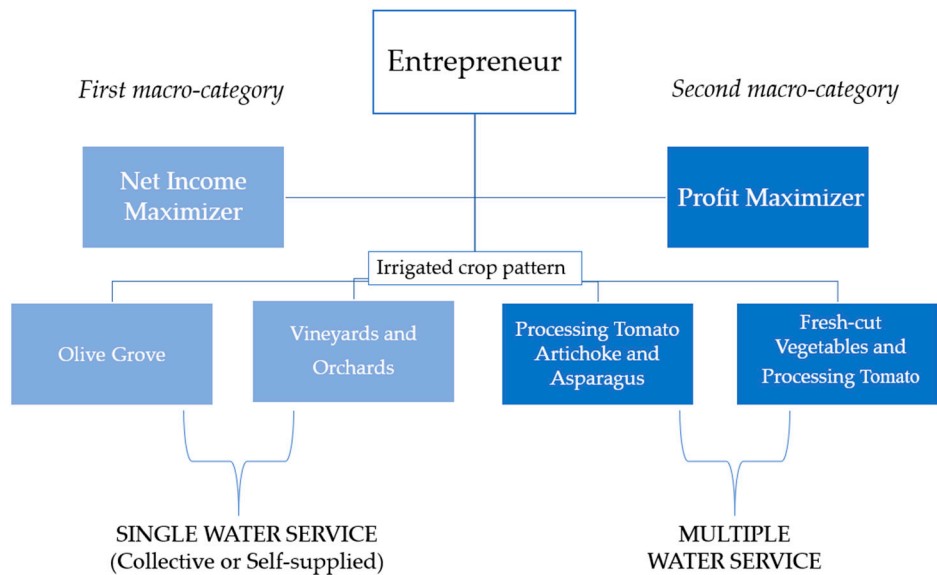

**Figure 2.** Cluster results representation. Source: our elaboration.

*3.2. Economic Value of Irrigation Water*

Table 5 indicates rental/purchase prices (€/ha) differentiated on the basis of the two entrepreneurial macro-categories.

**Table 5.** Rental/purchase land prices.

| | Rent (€/ha) | | Purchase (€/ha) | | Difference (Rain-Fed vs. Irrigated) | |
|---|---|---|---|---|---|---|
| | Non-Irrigated | Irrigated | Non-Irrigated | Irrigated | Rent | Purchase |
| *Net income maximizers* | | | | | | |
| Observations | 43 | 45 | 48 | 50 | - | - |
| Min | 100 | 300 | 12,000 | 20,000 | 200% | 67% |
| Max | 900 | 1500 | 46,000 | 67,000 | 67% | 46% |
| Mean | 425 | 1003 | 21,250 | 30,040 | 136% | 41% |
| SD | 238 | 291 | 5998 | 10,713 | - | - |
| *Profit maximizers* | | | | | | |
| Observations | 16 | 16 | 16 | 16 | - | - |
| Min | 200 | 1000 | 13,000 | 24,000 | 400% | 85% |
| Max | 800 | 1200 | 28,000 | 35,000 | 50% | 25% |
| Mean | 450 | 1025 | 20,562 | 28,500 | 128% | 39% |
| SD | 216 | 68 | 4049 | 3464 | - | - |

Source: our elaboration from direct survey.

For the macro-category of net income maximizers, there were 43 rent values for non-irrigated land and 48 purchase values for non-irrigated land. For irrigated land, on the other hand, 45 farmers declared a rent value and 50 a purchase value for land.

The average rent for non-irrigated land was between a minimum value of €100 and a maximum of €900 per hectare. Moreover, the average value amounted to 425.23 (€/ha).

Regarding irrigated land, the range was between €300 and €1500 per hectare with an average value of 1025 (€/ha). There was a percentage increase of 136% between the average rent values of land with and without irrigation.

Still considering the first macro-category, it can be observed that the purchase price of non-irrigated land fluctuated between €12,000 and €46,000 per hectare, with an average value of 21,250 (€/ha). Furthermore, the purchase price in cases of irrigated land was between €20,000 and €60,000 per hectare, with an average value of 30,040 (€/ha). In these cases, the percentage difference in the average values in the absence/presence of irrigation corresponded to 41%.

The same information was indicated for the second macro-category (profit maximizers). For this category, 16 observations were available for each type of value investigated.

Therefore, the average rent for non-irrigated land was between a minimum value of €200 and a maximum of €800 per hectare. Moreover, the average value was 450 (€/ha). For the rental values of land with the presence of water, minimum, maximum and average values corresponded to €1000, €1200 and €1025 per hectare, respectively. The percentage difference of the average rental value was 128%.

Lastly, considering the purchase value for non-irrigated land, the range was between €13,000 and €28,000/ha with an average value of €20,562/ha. The purchase value for irrigated land fluctuated between €24,000 and €35,000/ha with an average value of €28,500/ha. The presence of irrigation water involved a percentage increase in the purchase value of 39%.

The farmers' stated values for the land market in the sample were in line with results from the literature, confirming a higher value for irrigated land [17,26].

The mean value of water, expressed in m$^3$/ha, amounted to 1970 in the first macro-category and 3580 in the second (Table 6).

**Table 6.** Irrigation water volumes.

| Average Annual Unit Water Use (m$^3$/ha) | Net Income Maximizers | Profit Maximizers | Sample |
|---|---|---|---|
| Observations | 43 | 15 | 58 |
| Min | 466 | 844 | 466 |
| Max | 5000 | 6666 | 6666 |
| Mean | 1970 | 3580 | 2387 |
| SD | 1073 | 1939 | 1510 |

Source: our elaboration from the direct survey.

The difference between the annual rent for irrigated and non-irrigated land corresponds to the use value (Bi) obtained from irrigation. Furthermore, the difference between the purchase values reveals the capitalization value (Vi) of that benefit (Bi) in the long term [18,20]. Dividing each difference by the average irrigation volume, the monetary value of the water resource was obtained (€/m$^3$). The results are indicated in Table 7.

The results show a high similarity between macro-categories with regard to the land annual benefit (Bi). Indeed, the net income maximizers' benefit amounted to €578/ha against the €575/ha of the profit maximizers. Conversely, the capitalization values amounted to €8790/ha and €7938/ha, respectively.

On the other hand, the results relating to the water resource estimation value showed a greater difference between macro-categories. Indeed, the rent value corresponded to 0.29 (€/m$^3$) for net income maximizers and 0.16 for profit maximizers. Similarly, the purchase values were 4.46 and 2.21 (€/m$^3$), respectively.

**Table 7.** Water monetary value estimation.

| Entrepreneurial Figure | Rent | | Purchase | |
|---|---|---|---|---|
| | Δ Rent (Irrigated/Non-Irrigated) (Bi) (€) | Use value (€/m³) | Δ Purchase (Irrigated/Non-irrigated) (Vi) (€) | Capitalization Value (€/m³) |
| | (Bi) (€) | (€/m³⁾ | (Vi) (€) | |
| Net income maximizers | 578 | 0.29 | 8790 | 4.46 |
| Profit maximizers | 575 | 0.16 | 7938 | 2.21 |
| Sample | 578 | 0.24 | 8588 | 3.59 |

Source: our elaboration from the direct survey.

It seems that the differences in the estimated values for unit of volume depended on the structural (i.e., crop irrigation requirements) rather than the entrepreneurial characteristics of the two macro-categories, for which the total monetary values of irrigated land were equivalent.

*3.3. Evaluation of Irrigation Services (Collective or Self-Supplied)*

In this section, the results relating to the estimate of the water resource monetary value are reported, discriminating by type of water service (collective or self-supplied). Therefore, in this step, those observations with multiple services were excluded. Table 8 indicates the statistics of the rental/purchase land prices (€/ha).

**Table 8.** Descriptive statistics of rental/purchase land prices and water volumes.

| Water Service Type | Rent (€/ha) | | Purchase (€/ha) | | Average Water Cost (€/ha) | Average Annual Volumes (m³/ha) |
|---|---|---|---|---|---|---|
| | *Rain-Fed* | *Irrigated* | *Rain-Fed* | *Irrigated* | | |
| | | | Collective | | | |
| Observations | 13 | 13 | 15 | 17 | | |
| Min | 100 | 400 | 15,000 | 20,000 | 180 | 180 |
| Max | 800 | 1300 | 30,000 | 35,000 | 770 | 6250 |
| Mean | 450 | 1038 | 21,266 | 26,941 | 430 | 2605 |
| SD | 254 | 221 | 4620 | 5067 | 176 | 1516 |
| | | | Self-supplied | | | |
| Observations | 33 | 35 | 36 | 36 | | |
| Min | 100 | 300 | 12,000 | 20,000 | 50 | 584 |
| Max | 900 | 1500 | 46,000 | 67,000 | 1500 | 6000 |
| Mean | 435 | 1007 | 21,638 | 31,527 | 535 | 2071 |
| SD | 228 | 276 | 6432 | 11,721 | 408 | 1227 |

Source: our elaboration from the direct survey.

In the case of collective served land, rent vales for rain-fed land ranged from a minimum of €100 to a maximum of €800 per hectare. The average value amounted to 450 (€/ha). Rent values of irrigated land ranged between €400 and €1300 per hectare with an average of 1038 (€/ha). Regarding the purchase value, the range was between €15,000 and €30,000 per hectare for rain-fed land and between €20,000 and €35,000 per hectare for irrigated land. Moreover, the means of the purchase values were 21,266 and 26,461 (€/ha), respectively, for rain-fed and irrigated land.

In the case of farmland with self-supplied services, the rent value for rain-fed land ranged from a minimum value of €100 and a maximum of €900 per hectare. The average value amounted to 435 (€/ha). For irrigated land, minimum, maximum and average values corresponded to €300, €1500 and €1007 per hectare, respectively. Concerning the purchase values, the range was from €12,000 to €46,000 per hectare for rain-fed land and from €20,000 to €67,000 per hectare for irrigated land. The average values amounted to 21,638 and 31,527 €/ha for non-irrigated and irrigated land, respectively.

The average water cost was 430 €/ha for the group adopting the collective water service and 535 m$^3$/ha for self-supplied adopters. Unit cost for collective water use was obtained as the farmer's stated irrigation volume (m$^3$/ha) multiplied by the volumetric three-tier water tariff scheme applied by the Reclamation and Irrigation Board of Capitanata (0.12 €/m$^3$ with a unit volume under 2050 m$^3$/ha; 0.18 €/m$^3$ with a unit volume ranging from 2051 to 3000 m$^3$/ha; 0.24 €/m$^3$ for further amounts), plus a yearly fixed component of 60 €/ha. In the case of self-supplied services, the farmer's stated value of the annual total cost for energy (either electricity or fuel) was divided by the irrigated area. The initial investment cost for the pumping system was not accounted for, being viewed as a sunk cost.

The average annual volumes used were 2605 m$^3$/ha and 2071 m$^3$/ha, respectively.

Table 9 shows the monetary unit values of water resources, differentiating by water service type. Furthermore, by applying Equation (3), the capitalization rate was obtained.

**Table 9.** Water resource monetary value estimation and capitalization rate.

| Water Service | Rent Value | | Purchase Vale | | Capitalization Rate |
|---|---|---|---|---|---|
| | Δ Rent (Irrigated/Rain-Fed) (Bi) (€) | Rent value (€/m$^3$) | Δ Purchase (Irrigated/Rain-Fed) (Vi) (€) | Purchase Value (€/m$^3$) | |
| Collective | 588 | 0.22 | 5675 | 2.18 | 0.10 |
| Self-supplied | 572 | 0.27 | 9889 | 4.77 | 0.06 |
| Sample | 578 | 0.24 | 8588 | 3.59 | 0.07 |

Source: our elaboration from the direct survey.

It emerges that there was little difference in monetary value in the short term (rent value). Indeed, for the collective and self-supplied services, the magnitude of the monetary rent value was equivalent. Conversely, in the long run, farmers considered the self-supplied service to be more valuable. The difference in value in the land sales market between land served by collective and self-supplied irrigation water amounted to €4214/ha, regardless of all other land characteristics. This result translates into a capitalization rate of 0.10 for the collective and 0.06 for the self-supplied services. This finding proves that the self-supplied service from groundwater is much more reliable than collective services based on regulated surface water.

## 4. Discussion

In this study, the economic value of irrigation services was assessed. By comparing land sales values of irrigated and non-irrigated farmland, the unique value of irrigation water was calculated. In addition, differences between self-supplied and collective irrigation services were verified. The findings indicated higher capitalization values in the case of lands served by self-supplied water from groundwater sources. Indeed, in the long term, a greater degree of reliability was found for the self-supplied rather than the collective services. Such a difference reflects the economic border of uncertainty in the irrigation services provided by the Reclamation and Irrigation Board of Capitanata. Giannoccaro et al. [24] found a fall of 30% in farmer revenue when the collective service of Capitanata cut down the water delivery. Akbaya [27] identified 5.76% as the capitalization rate for

irrigated areas in Turkey. Giannoccaro et al. [20], in the Capitanata area, found an average rate of 7%, while a rate of 8% was indicated by Mesa-Jurado [28] in Southern Spain. In this study, the average capitalization rate was 7%. However, significant differences of up to four percentage points were found between the self-supplied and collective services, with the latter showing the higher rate. Therefore, the initial hypothesis, that the land market would be able to reveal the greater reliability self-supplied groundwater, has been proven. Joshi et al. [17], in Nepal, found that farmers with access to different sources of irrigation water attributed a different value to the water resource.

These findings are also in line with the recently published results of La Sala et al. [29] for the Capitanata area. They found that, for winegrowing farms, the most efficient irrigation source was a private well, followed by the simultaneous presence of a well and a collective network, and lastly a network alone. Nevertheless, they inaccurately indicated that groundwater use is almost free (except for a small, fixed fee paid every five years) while irrigation water from the collective network is more expensive. Actually, they neglected the running cost of self-supplied groundwater for irrigation, which was accurately reported in our research. In doing so, they formulated misleading policy implications, following the mainstream literature (e.g., banning licensing or taxation for groundwater use), with regard to achieving sustainable water management.

The economic evaluation of water services for irrigation should steer policy makers and water use associations in setting suitable pricing policies aimed at the enhancement of water use efficiency. According to the estimates, the value of irrigation water revealed from the land market ranges from 0.22 to 0.27 €/m$^3$. Such a value is actually the economic rent (not be confused with income from renting out property); that is, the payment a factor generates in excess of the cost of using it. As a consequence, a rise in the price of water for irrigation reduces the farmers' economic rent without making any reduction in water demand. A review of public evidence and scientific research regarding the effect of pricing on irrigation water demand shows the limitations of water pricing and the need to integrate pricing and non-pricing instruments [30]. Furthermore, Portoghese et al. [12], in the case of conjunctive use of water services, have pointed out the relevance of the interplay between pricing policy applied for collective facilities and groundwater use, with a detrimental side effect on groundwater overextraction.

Based on the findings, in order to improve the reliability of the water volume delivered by collective services, the Reclamation and Irrigation Board of Capitanata should consider their water rights allotment based on a volume guarantee. Priority rights could have a higher degree of reliability, associated with a higher price [8]. In doing so, collective services could apply price discrimination based on the guarantee of water delivery. There would be some farmers willing to pay more for ensuring irrigation services. Although it would be unknown who would pay more, water user associations of collective services could offer the farmers a menu of different options (and prices) so that they reveal their willingness to pay by selecting a particular option. A more flexible water rights system would improve allocation efficiency among farmers with different crop patterns (e.g., winegrowing vs. arable crops).

The findings reported in this research should be considered as explorative and further research needs to be carried on.

**Author Contributions:** Conceptualization, G.G.; methodology, G.G.; validation, L.M., B.C.d.G. and G.G.; formal analysis, L.M.; data curation, L.M.; writing—original draft preparation, L.M.; writing—review and editing, G.G.; supervision, G.G.; project administration, G.G.; funding acquisition, B.C.d.G. All authors have read and agreed to the published version of the manuscript.

**Funding:** Survey of farmers was carried out within the project "Economia delle risorse irrigue in Puglia", CUPB37G17000030007, funded by the Regional Government of Apulia.

**Institutional Review Board Statement:** The study was conducted according to the guidelines of the Declaration of Helsinki.

**Informed Consent Statement:** Informed consent was obtained from all subjects involved in the study.

**Data Availability Statement:** The data presented in this study are available on request from the corresponding author. The data are not publicly available due to the national law on the restriction of privacy.

**Acknowledgments:** We are thankful for insights received from the editor and three anonymous reviewers. We thank the University of Bari Aldo Moro for the research fund program "Fondi a supporto della ricerca", 2015–2016, due to which in-depth English editing was possible.

**Conflicts of Interest:** The authors declare no conflict of interest. The funders had no role in the design of the study; in the collection, analyses, or interpretation of data; in the writing of the manuscript, or in the decision to publish the results.

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
