# Peer review of "Farmer Evaluation of Irrigation Services. Collective or Self-Supplied?"

_land, doi:10.3390/land10040415_

Round 1

Reviewer 1 Report

Dear authors, The work has been prepared correctly. However, I have a few remarks that I made at work, in the comments. I would suggest rewriting the work by including data sources in the materials and methods section first, because the questions that arise are explained in this subsection. There are also some inaccuracies and some simplifications in the work, so I would also like to ask for clarification. However, the work is interesting and valuable and after taking into account the comments, I recommend publishing it. 

Author Response

We are very grateful for your comments and useful suggestion.

Hereafter, point-to-point replay to given suggestion.

Many thanks.

 All editing suggestions have been enclosed in the main text.

Comment: -What part is this group of farmers? I mean, what is the percentage of all farmers or landfarms? Why did the authors not use e.g. FADN data. Then the results could be compared with other countries.

Response: Ok, thank you. We integrated the main text with some further details -see Table 2. Moreover, in the map is now highlighted (Figure 1). Using data from FADN will able us for comparison. Unfortunately, FADN do not report leasing values and other specific data about the management of water resource (e.g. presence of on-farm metering device or adoption of digitalised irrigation management systems). Furthermore, data set of FADN has been cut down. For a very focused area such as our study area, FADN accounts for very few number of farms. By contrast, by collecting data by direct survey we obtained information about rent and purchase land values from as much as respondents. Such a good point has been stressed in the main text (line 247-250).

Comment: How to understand this division:

For example: water services- 3 both services=Multiple water services? Why are the remaining coefficients missing?

Response: Thank you. There was a missing value. Moreover, we now report for water services % values.

Comment: Lack of data. What kind of innovations authors mean?

Response: Ok, thank you. Kind of innovation is specified (Table 1).

Comment: Why these data weren't used as a variables in cluster analysis?

Response: Actually, variables such as soil fertility, distance from nearest town and road, slope and size of each farm plot were not used. In fact, we were focused on the entrepreneurship nature of farmers. The farmland size, both own and rent-in lands were used in clustering. Moreover, cluster analysis did not result in other significant variables such as slope and so on. This could be for small numbers of cases (sample size) regressed against larger number of variables.  Finally, it is worth mentioning that the study area shows a high homogenous land structure (flat area, medium and large farm size, located in rural area with very low inhabitant’s rate).

Comment: It would be good to clearly indicate the division into macro categories here.

Response: Yes, thank you. We reshaped Figure 2 in order to make clearer the division into macro-categories. Also some details have been added in the main text reporting details on what difference between two macro-categories (line 302-319).

Comment: How this information about prices relates to that posted below [from line 267 to  270)? The point is that the cost of the plot cannot depend on the approach to the macro category. This is evidenced by the similar cost values in both categories. It must be borne in mind that in both groups different farms with different economic approaches are taken into account. The proposed division in this context is an oversimplification.

Response: Sale comparison approach bases on the assumption of “all other things invariant”. Thus, cluster analysis was run at the beginning with the aim of grouping cases into homogenous sub-sets. In fact, crop pattern, especially irrigated corps are the main difference that cluster analysis reported. As a matter of fact, net income maximizer exhibits smaller farmland size and is basically specialized in permanent crops (i.e. orchards, vineyards and olive groves). By contrast, profit maximizer relies on larger farmland size and is oriented in arable crops (i.e. vegetables and processing tomato). A further important differentiation between the two macro-categories is the type of water service adopted. In particular, the net income maximisers indicate almost exclusively a single water service (collective or self-supplied), conversely, profit maximisers rely on multiple irrigation services.

In this way, we excluded that differences in the land market values are related to two macro-categories. Indeed, it seems that the difference in the estimated values for unit of volume depends on the structural (i.e. crop irrigation requirements) rather than the entrepreneurial characteristics of the two macro categories, for which total monetary values of irrigated land are equivalent (table 7). Then, the subsequent analysis of Evaluation of irrigation services (collective or self-supplied) has been performed without differentiating for entrepreneurial figures. Therefore, in both groups (collective vs. self-supplied) different farms with different economic approaches have been taken into account.  

Comment: This is not just a test result, but a simple statement. It is known that irrigation infrastructure and access to water represent an additional investment cost that will increase the value of the land.

Response: Ok, we rephrase this statement. It is worth mentioning the fact that our data was self-reported by farmers. Such results, give evidence of data reliability.

A .pdf file is attached with tracks of all changes made. 

Reviewer 2 Report

Review on the manuscript: Farmer evaluation of irrigation services. Collective or self-supplied?

A well written paper on a hedonic price evaluation for different types of farmland irrigation distinguishing ownership types and water supply – collective or self-supplied. The method is well explained and appropriate, some details need to be better explained. The results are well described and discussed, although not properly structured (see comments). After some minor revisions I suggest publication.

Some more comments and corrections below.

P1L31 in the European Union

P2L58  … rivers

P2L78ff the last two paragraphs would rather fit into section 2. Just mention the method briefly in clearly stated objectives. It the moment, they are somehow hidden in L73 ff.

P3L96 possible to obtain

P3 equations: explain r

P8 figure 2: please use the same terms as in the describing text. You mention net revenue in the text, net margin in the figure. Please also explain the meaning of your categories. Does the first group just want a decent income, but might not observe whether their capital is diminishing?

P11 table 7: maybe it is explained earlier, but it would be helpful to have the information on how the water costs per ha are calculated. Either in the figure caption or in the text.

P12L340 typo: cut down

Conclusions should not contain any more references, but only conclude the findings. This section is mainly a discussion. I suggest to have a proper discussion section with a deeper discussion of the results comparing to other literature. The last parts in the results section also belong to discussions. Also reflect on the implication of the value differences for policies and organizations.

P12L361 As suggested above your way of calculating the water cost is not clear. Please explain better above what is included in this term.

Add some proper conclusions summarizing your findings.

Author Response

We are very grateful for your comments and useful suggestion.

Hereafter, point-to-point replay to given suggestion.

Many thanks.

All editing suggestions have been enclosed in the main text.

Comment: P3 equations: explain r

Response: Ok, made in the main text.

Comment: P8 figure 2: please use the same terms as in the describing text. You mention net revenue in the text, net margin in the figure. Please also explain the meaning of your categories. Does the first group just want a decent income, but might not observe whether their capital is diminishing?

Response: Thank you, we uniformed the word in the text as well as in the figure. We also added further details in the text about categories (see from line 306 to 319). According to the results, farmers belonging to the first category are assimilated to net income maximazer, preferring a familiar management system, and providing themselves a larger share of the capital and the labour. They do not account for opportunity cost of own endowments but only the explicit costs. Whereas the second macro-category of farmers, acquiring almost all inputs directly on the market, is interested in maximizing its profit and prefer a more professional management system.

Comment: P11 table 7: maybe it is explained earlier, but it would be helpful to have the information on how the water costs per ha are calculated. Either in the figure caption or in the text.

Response: Thank you for your suggestion. Through direct survey we obtain information about the costs borne by farmers for water service both collective and self-supplied. To obtain the average unit cost we multiplied farmer’s stated irrigation volume by volumetric three-tier water tariff scheme applied by Irrigation and Reclamation Board of Capitanata. In the case of self-supplied service, farmer’s stated value of annual total cost for energy (either electricity or fuel) is divided by irrigated area. As you suggested we reported this further information in the text (see from line 407 to 415).

Comment: Conclusions should not contain any more references, but only conclude the findings. This section is mainly a discussion. I suggest to have a proper discussion section with a deeper discussion of the results comparing to other literature. The last parts in the results section also belong to discussions. Also reflect on the implication of the value differences for policies and organizations.

Response: Ok thank you. We adopted the division of the article section as you suggested.

Comment: P12L361 As suggested above your way of calculating the water cost is not clear. Please explain better above what is included in this term.

Response: Thank you. Bearing in mind the previous comment we hope to have made this term clearer (see from line 407 to 415).

Comment: Add some proper conclusions summarizing your findings.

Response: We ended section of Discussion with proper concluding remarks. In line with editorial settlement of LAND journal and as suggested by other reviewers, conclusion section is not due.

A .pdf file is attached with tracks of all changes made. 

Reviewer 3 Report

Introduction

  • Line 42- the author’s mentioned “drought”- what kind of drought dealing in this context ???
  • The author’s failed to explain –“why the paper deserves to publish? What is the contribution to the knowledge or literature?

Method

  • Before specifying the equation – the author’s need to have one section about the conceptual /theoretical framework of the mode (Hedonic pricing model- sale comparison approach)

Please refer to the theoretical framework of the hedonic model :

Owusu, V., E. Sekyere, E. Donkor, N.A. Darkwaah, and D. Adomako-Boateng. 2017. Consumer perceptions and willingness to pay for cassava-wheat composite bread in Ghana: A hedonic pricing approach. Journal of Agribusiness in Developing and Emerging Economies 7: 115–34.
  • Hedonic approach -applied the price attribute of the characteristics of valuing water- therefore. The model specification should be clear with the dependent variable ( value of waters against explanatory variables o (attribute of the characteristics)- unless applying hedonic model does not help
  • Line 103-105- the equation number should specify
  • Line 158- need a source for figure 1
  • Line 140- sampling technique missing- why you choice Apulia- why not other areas- what is the reasoning of selection- you should convince scientifically – for example using multi-stage- randomly and balloon and so on
  • Line 160- how you reach 75 observation – why not 80- you need to convince scientifically – whether you use random sampling formula and so on – please refer to Cochran’s (1977)
  • Line 238- source for figure 2

Result and discussion

  • If I not mistaken, according to the “land” journal guideline – result and discussion should be separated

Conclusion

  • I do not think so, and it is necessary to include citations in the conclusion section “Joshi et al. [13]” and others …..

Author Response

We are very grateful for your comments and useful suggestion.

Hereafter, point-to-point replay to given suggestion.

Many thanks.

All editing suggestions have been enclosed in the main text.

Comment: Line 42- the author’s mentioned “drought”- what kind of drought dealing in this context ???

Response: Thank you for your comment. As you suggested we specify that we are referring to hydrological drought. Unlike the meteorological, hydrological drought occurs when low water supply becomes evident, especially in streams, reservoirs, and groundwater levels, usually after many months of meteorological drought. This is the case of this work and the one mentioned.

Comment: The author’s failed to explain –“why the paper deserves to publish? What is the contribution to the knowledge or literature?

Response: Thank you for your point of view. We have strengthened this aspect throughout the text, mostly in the section of Introduction.

Comment: Before specifying the equation – the author’s need to have one section about the conceptual /theoretical framework of the mode (Hedonic pricing model- sale comparison approach)

Response: Thank you. We give more information about the theoretical framework  (see from line 115).

Comment: Please refer to the theoretical framework of the hedonic model :

Owusu, V., E. Sekyere, E. Donkor, N.A. Darkwaah, and D. Adomako-Boateng. 2017. Consumer perceptions and willingness to pay for cassava-wheat composite bread in Ghana: A hedonic pricing approach. Journal of Agribusiness in Developing and Emerging Economies 7: 115–34.

Response: Thank you, we treasured from the work you suggested adding suggested reference (see from line 115).

Comment: Hedonic approach -applied the price attribute of the characteristics of valuing water- therefore. The model specification should be clear with the dependent variable ( value of waters against explanatory variables o (attribute of the characteristics)- unless applying hedonic model does not help

Response: Sale comparison approach bases on the assumption of “all other things invariant”. Thus, cluster analysis was run at the beginning with the aim of grouping cases into homogenous sub-sets. In fact, crop pattern, especially irrigated corps are the main difference that cluster analysis reported. As a matter of fact, net income maximizer exhibits smaller farmland size and is basically specialized in permanent crops (i.e. orchards, vineyards and olive groves). By contrast, profit maximizer relies on larger farmland size and is oriented in arable crops (i.e. vegetables and processing tomato). A further important differentiation between the two macro-categories is the type of water service adopted. In particular, the net income maximisers indicate almost exclusively a single water service (collective or self-supplied), conversely, profit maximisers rely on multiple irrigation services.

In this way, we excluded that differences in the land market values are related to two macro-categories. Indeed, it seems that the difference in the estimated values for unit of volume depends on the structural (i.e. crop irrigation requirements) rather than the entrepreneurial characteristics of the two macro categories, for which total monetary values of irrigated land are equivalent (table 7). Then, the subsequent analysis of Evaluation of irrigation services (collective or self-supplied) has been performed without differentiating for entrepreneurial figures.

Variables such as soil fertility, distance from nearest town and road, slope and size of each farm plot did not report significance in cluster analysis. This could be for small numbers of cases (sample size) regressed against larger number of variables. Finally, it is worth mentioning that the study area shows a high homogenous land structure (flat area, medium and large farm size, located in rural area with very low inhabitant’s rate).

Comment: Line 140- sampling technique missing- why you choice Apulia- why not other areas- what is the reasoning of selection- you should convince scientifically – for example using multi-stage- randomly and balloon and so on

Response: The area is chosen as explorative case study, in which conjunctive use of surface and groundwater resources has been active from longer (see Introduction).

A snowball sampling procedure was followed in order to align as much as possible sampled cases to irrigated crop pattern, annual water used and irrigation services type of study area. Sampling was conducted in a homogenous area with relevant cases of conjunctive use of both irrigation services (see red circle in the Figure 1). Despite the size of the sample, a higher proportion of farmland area of the study area was sampled, namely 4,861 hectares of total utilized agricultural area with 2,017 ha actually irrigated. As a whole, sample reports a good representativeness of irrigated crops (Table 2), by water service type (Table 3) as well. Details on this point have been added into the main text (line 209-2016). Table 2 has been also added.

Comment: Line 160- how you reach 75 observation – why not 80- you need to convince scientifically – whether you use random sampling formula and so on – please refer to Cochran’s (1977)

Response: Cochran’s (1977) is an interesting reference but somewhat aged. Sampling techniques in social science (economics) have evolved and case study approach has been increasingly drawing the attention of scientists. It is worth mentioning that the study area shows a high homogenous land structure (flat area, medium and large farm size, located in rural area with very low inhabitant’s rate). Actually, the size of sample was constrained to budget availability.  We clearly stated in the main text the explorative nature of our sample while a final disclaimer has been also enclosed in the section of discussion.

Comment: If I not mistaken, according to the “land” journal guideline – result and discussion should be separated

Response: Ok, thank you. We amended it.

Comment: I do not think so, and it is necessary to include citations in the conclusion section “Joshi et al. [13]” and others

Response: Thank you. We ended section of Discussion with proper concluding remarks. In line with editorial settlement of LAND journal and as suggested by other reviewers, conclusion section is not due.

A .pdf file is attached with tracks of all changes made. 

Round 2

Reviewer 1 Report

Dear Authors,

After reading the revised version of the article, I think that it has improved significantly. Thank you for responding to my remarks and comments. I believe that this article can be published in current form.

Reviewer 3 Report

The autor/s amend all my concerns such as type of drought; contributions to knowledge; conceptual/theoretical framework, sampling technique, and others.